# A Linked Data Application Framework
# to Enable Rapid Prototyping

Markus Schröder, Christian Jilek, and Andreas Dengel

[1] Smart Data & Knowledge Services Dept., DFKI GmbH, Kaiserslautern, Germany
[2] Computer Science Dept., TU Kaiserslautern, Germany
{markus.schroeder, christian.jilek, andreas.dengel}@dfki.de

**Abstract.** Application developers, in our experience, tend to hesitate when dealing with linked data technologies. To reduce their initial hurdle and enable rapid prototyping, we propose in this paper a framework for building linked data applications. Our approach especially considers the participation of web developers and non-technical users without much prior knowledge about linked data concepts. Web developers are supported with bidirectional RDF to JSON conversions and suitable CRUD endpoints. Non-technical users can browse websites generated from JSON data by means of a template language. A prototypical open source implementation demonstrates its capabilities.

**Keywords:** Linked Data · RDF · JSON · Converter · Platform

## 1 Introduction and Motivation

The approach of linking data enjoys great popularity in our projects. However, we observe that the adoption of linked data, especially for beginners, does not show a flat learning curve because of rich technology stacks, standards and new concepts that need to be studied first. This includes, for instance, the Semantic Web layer cake, RDF basics (statements, literals), ontologies, SPARQL and so on. The multitude of technologies can be overwhelming for application developers who come in contact with them for the first time. For these reasons, our experience has shown that they tend to hesitate when dealing with such technologies, especially in the industrial sector. A suitable development framework could reduce the initial hurdle to build linked data applications and eventually work with linked data.

Once such data is introduced in a project, various users despite their different roles should be able to participate and contribute to it. That is why we think that linked data applications should be designed to consider at least these major user groups: Semantic Web practitioners, web developers and non-technical users such as knowledge workers. Because linked data is already stored in a desired data model for Semantic Web experts (typically RDF), they can directly work with it, for example by using SPARQL queries, without data conversion needs. In contrast, web developers, not necessarily experienced with Semantic Web technologies, in our opinion rather expect a JSON-based API to be available in

such systems which may also support basic Create, Read, Update and Delete operations (CRUD). Furthermore, non-technical users usually have a demand for graphical user interfaces (GUI), in our projects often HTML and JavaScript-driven web apps, that provide various forms to let them browse and manipulate data dynamically.

We propose a Linked Data Application Framework (LDAF) in which these requirements are recognized. Our design decisions are outlined in the following. Such a framework should ensure that managed hyperlinks (URIs) are resolvable which is essential in linked data. Links are a known concept for users and enable the querying, referring and ultimately linking of resources. Even non-technical users are familiar with URLs due to their browsing in the World Wide Web. However, when links are surfed, users usually expect to receive data in a familiar format. This is typically accomplished with content negotiation, however, it requires that linked data has to be properly converted to JSON for web developers and HTML for non-technical users. Therefore, our proposal includes that such applications should provide a way to bidirectionally transform between JSON and RDF. By additionally providing CRUD operations with POST, GET, PUT, PATCH and DELETE request methods, web developers are able to build services with a JSON-based API that sufficiently retrieve, manipulate and especially link resources. To render well-designed HTML pages for knowledge workers, a template language can be utilized that is fed with JSON object representations just mentioned. Such websites can also make use of asynchronous HTTP requests to build dynamic web apps. Besides simple CRUD resources, the framework should also be extendable with further linked data resources to satisfy more complex use cases.

We see in the usage of LDAF several opportunities. With almost no additional effort Semantic Web enthusiasts are able to provide linked data applications for their project partners not acquainted with Semantic Web standards. With that, the group can directly make use of the evolving linked data graph while contributors entering and manipulating data on websites or through programmed web services. This rapid prototyping can push early discussions about the modeled domain in projects and enable developers to test their first knowledge services, like for instance knowledge assistants[3]. Furthermore, it could serve as a initial basis for discussions when constructed knowledge graphs need to be reviewed by users.

In this paper, we present our first prototypical implementation of the envisioned framework.

## 2   Demo

Our open source implementation is written in Java and hosted on GitHub[4]. The example application from our tutorial is available on a demo page[5] for testing. What follows are implementation details of its features.

---

[3] `https://comem.ai/SensAI`

[4] `https://github.com/mschroeder-github/ldaf`

[5] `http://www.dfki.uni-kl.de/~mschroeder/demo/ldaf`

With multiple independent users in mind, our framework implements basic registration and login functionalities. Each registered user is associated with a private RDF graph that stores the person's personal information like names and password (hash). However, this does not restrict application builders to create shared graphs for user groups. Signed-in agents have access to the subsequently described resources.

To ease the development, our framework already provides several default implementations of commonly required resources. The `/ontology` resource serves the read-only terminology and makes sure that links to concepts are always resolvable. A `/search` resource provides a simple SPARQL and regular expression based search in resource labels. The `/sparql` endpoint lets experts run self-written SPARQL queries on their visible RDF graphs. An image uploader at `/upload` allows users to send and link depictions of their resources. If the default implementations do not fit a user's needs, our framework is easily extensible with Java classes extending `LinkedDataResource`. All resources perform content negotiations together with format conversions to correctly respond to the MIME types `text/html`, `text/turtle` and `application/json`.

To accomplish the latter, a `Converter` is provided that bidirectionally transforms RDF statements to JSON object representations. Given a starting resource, the RDF graph is traversed in a configurable depth in order to create for each resource nested JSON objects containing the resource's `uri`, `path`, `localname` and further properties as keys. A special `_incoming` object lists all properties and subjects that refer to it. Our converter tries to achieve a good trade-off between RDF(S) expressiveness and a more lightweight JSON representation that provides an intuitive object view for web developers not familiar with Semantic Web concepts.

To provide HTML pages, we make use of the template language Apache FreeMarker[6]. Using the converted JSON objects for the data model, equivalent representations can be built with HTML, CSS and JavaScript. Additional forms and asynchronous HTTP requests let non-technical users view and manipulate linked data dynamically.

To archive this, our framework also provides default implementations for CRUD operations and pagination. Every component, especially the Create component, ensures that all URIs managed by the application are always resolvable.

## 3   Related Work

The related Linked Data Platform[7] also describes how to read and write linked data with HTTP operations. However, this standard mainly considers RDF as an exchange format, while other formats (HTTP, JSON) are only marginally mentioned (see Section 6.2.1).

SPARQL Template [1] is a related approach to generate other formats from RDF. It may also be used to build JSON objects or HTML pages, but it requires

---

[6] `https://freemarker.apache.org`
[7] `https://www.w3.org/TR/ldp`

sufficient experience in SPARQL. Similarly, one could easily provide JSON-LD[8] when JSON data is desired. Yet, we did not consider this RDF serialization format, since it needs additional training to comprehend its 23 @-notations (like `@context`) and its sometimes unexpected nesting.

In former work, we already demonstrated how Semantic Web newcomers are enabled to interact with associated technologies: A path based JSON REST API allows users to perform CRUD operations on RDF graphs [2], while a generator is able to build relational databases and REST APIs from RDF [3]. This work ties together previous ideas and new ones in a dedicated framework.

## 4   Conclusion and Outlook

Because of our observation that people often tend to hesitate when dealing with Semantic Web technologies due to its complexity, we proposed a Linked Data Application Framework (LDAF) with the intention to reduce initial hurdles and enable rapid prototyping, especially for web developers and non-technical users. A prototypical open source implementation demonstrated its capabilities and possible opportunities.

In the future, we plan to deploy our framework in projects with different domains and users. A major use case will be the integration of domain experts as Humans-in-the-Loop (HumL) during the construction of knowledge graphs. For that, we intend to provide in future versions of our prototype an HTML-GUI generation approach comparable with [3].

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
