# OpenReview forum: "A Linked Data Application Framework to Enable Rapid Prototyping"
_eswc-conferences.org/ESWC/2021/Conference/Poster_and_Demo_Track — Submitted to ESWC2021 P&D_

### Official Review · AnonReviewer1 · 2021-04-13
**Interesting tool, should have better compared with other works**

**Rating:** 5
**Confidence:** 5

**Review:**

The paper describes the LDAF framework for accessing semantic web resources by SPARQL/rdf, json (for web developers), HTML (for final users). The authors provide the link to a demo and to a well-documented repository.

The presented application is interesting and relevant to the conference. It is unclear from the paper how much of the API/HTML setup is automatic (coming from the analysis of the graph) or manual (configuration files).

The authors' claim about barriers avoiding developers to use LD should be supported with references such as https://hal.inria.fr/hal-02058664v2 or https://github.com/w3c/EasierRDF .

The related work session is quite poor. Two papers from the same authors are included, but it is unclear if these work are somehow re-used/applied in LDAF. Moreover, there is quite important literature in generating both Web APIs (in JSON) and HTML visualisations on LD. I believe that some relevant examples could have been included and compared.

**Anonymity:**

Yes, I would like my review to remain anonymous.

---

### Official Review · AnonReviewer2 · 2021-04-14
**Interesting to broad the SWeb technologies audience**

**Rating:** 6
**Confidence:** 4

**Review:**

The authors propose a framework to reduce the adoption barrier of consuming Linked Data in your applications due to the learning curve that the Semantic Web stack imposes.
It basically provides web developers and end-users with simplified interaction methods (a JSON-based API for CRUD ops for the former ones, and a template based visualization for the latter ones). It's an interesting proposal to bring together different stakeholders involved in the development of the app, otherwise, in my experience, it can be quite complicated to coordinate the different teams as the data model evolves (following the description, it would be interesting as well only for knowledge engineers working in a collaborative way).

Some questions that arise:
- how new schemas/ontologies/vocabularies and their evolutions are handled to provide the JSON representation? and the template generation? Building static mappings might go against precisely the evolution of the Linked Data source; breaking the app, and therefore hampering the perception of utility and usability of Linked Data amongst the newcomers.
- Regarding the prototype, it seems that only local triplestores can be used; I haven't been able to find a configuration for accessing precisely third-party repositories.
- I've tried to try the demo, but it was giving a 404 when I tried to log as admin, and after registering ...
- The authors have discarded JSON-LD arguing that it requires comprehending the different annotations it has; however, I fail to see the benefits of leaving free JSON "in the wild" especially if the knowledge model is complicated. Is there anyway to control the complexity of the generated JSONs or is something that is completely configured (in this case) by the knowledge engineering stuff to be understood by the web developers? I would have tried it myself, but as said above, the demo seems to be broken.

Typo:
page 3: other formats (HTTP, JSON) => shouldn't it be HTML?

**Anonymity:**

Yes, I would like my review to remain anonymous.

---

### Official Review · AnonReviewer3 · 2021-04-18
**good intensions to promote Linked Data adoption but no clear innovation**

**Rating:** 5
**Confidence:** 4

**Review:**

This is a demo paper about the Linked Data Application Framework (LDAF). LDAF is proposed to abstract the peculiarities of semantic Web technologies for application developers which hesitate to use Semantic Web technologies.

Indeed the adoption of Semantic Web technologies by application developers is a topic that has concerned the semantic web community and different solutions were proposed, including the ones the authors mention in their related work section. In this respect, the proposed demo is in the right direction.

However, there are a few aspects of this paper that could be improved in relation to its basis and innovative aspects, related work and presentation style.

As far as related work is concerned, the paper refers to previous work of the same authors but it does not thoroughly touches the major directions of previous work aiming to simplify access to Linked Data via APIs. SPARQL Template and the Linked Data Platform are mentioned but there are many more works in this, such as the NGSI-LD (https://forge.etsi.org/rep/NGSI-LD/NGSI-LD/tree/master), the Linked Data API specification by UK Gov (https://github.com/UKGovLD/linked-data-api/blob/wiki/Specification.md), or even LDflex (https://ldflex.github.io/LDflex/).

My question would then be what are the innovative aspects of the solution of this demo? How is it different compared to the state of the art?

The option of using JSON-LD is disregarded because JSON-LD has the @-notations that the developers need to be aware of. However, the @-notations do not necessarily need to be embedded in the JSON document, they can be all hidden in the @context which can also be externally referenced with a URI (https://www.w3.org/TR/json-ld11/#the-context). If the latter occurs, then there is no distraction for the application developers but also, if there is an embedded context, that can be ignored by the application developers. Regarding the issue of unexpected nesting, I would need an example to understand what is meant.

Last, regarding the bidirectional transformation, I see the value of having such transformations, but I am not sure about the benefit of its use in this content. Why would one need to transform the JSON data to RDF statements and then again to JSON to make them accessible? Why the intermediate RDF statements are needed if in the end the semantic annotations are disregarded? Wouldn't it be easier to just make the JSON data directly available? Maybe there is a good motivation but it is not clearly described in the current version of the paper.

In the long term, it would be interesting to have a user study on the impact of such solutions have to the users, do they actually lower the barriers for Linked Data adoption.

**Anonymity:**

Yes, I would like my review to remain anonymous.

---

### Official Review · Program_Chairs · 2021-04-18
**Metareview: Reject (Lack of context regarding related work, unclear design choices)**

**Rating:** 5
**Confidence:** 5

**Review:**

This was a borderline paper, with two reviewers leaning reject, and one leaning accept. In general, there were concerns about the lack of discussion of related work, which left the novelty of the current proposal unclear. There were also concerns about some of the design choices, such as unclear reasons for discarding JSON-LD as an option. For these reasons, we feel that the paper is not ready for publication at this time.

**Anonymity:**

Yes, I would like my review to remain anonymous.

---

### Decision · Program_Chairs · 2021-04-19

Reject